# Sustainability Meets Information Technologies: Recent Developments and Future Perspectives

**Andrés Felipe Valderrama Pineda [1],*, Iva Ridjan Skov [1], Hanaa Dahy [1], Jamal Jokar Arsanjani [1], Ida Maria Bonnevie [1], Tom Børsen [1] and Maurizio Teli [2]**

[1] Department of Sustainability and Planning, The Technical Faculty of IT and Design, Aalborg University, 2450 Copenhagen, Denmark; iva@plan.aau.dk (I.R.S.); hanaadahy@plan.aau.dk (H.D.); jja@plan.aau.dk (J.J.A.); idarei@plan.aau.dk (I.M.B.); boersen@plan.aau.dk (T.B.)

[2] Department of Sustainability and Planning, The Technical Faculty of IT and Design, Aalborg University, 9000 Aalborg, Denmark; maurizio@plan.aau.dk

*   Correspondence: afvp@plan.aau.dk

**Abstract:** This article aims at addressing the future challenges in Sustainability and Information Technology (IT) by reversing the order of the conventional prioritization of social objectives and technology, and placing the aim first and the means second. In engineering and technology, historically, there has been greater focus on first developing the technologies (means) and then determining their potential (aim), and how to tame their unintended consequences. The greatest challenge confronting humanity in the coming decades is sustainability. Therefore, the question is how can IT design, develop, and assist in maintaining the ambitious, albeit difficult to grasp, sustainability agenda? This discussion is pertinent in order to avoid research programs and academic curriculum which dive into the intricacies of IT without viewing sustainability as a core value, which ultimately risks replicating the historical pattern that will generate even more unsustainability.

**Keywords:** sustainability and IT; research; education

## 1. Introduction

The Technical Faculty of IT and Design at Aalborg University (AAU) is currently developing an ambitious strategic project to create more IT-centered educational programs. The motivation is to address the current and projected needs of IT experts in all areas in Denmark, including public administration, industry, civil society, and infrastructure. This project resonates with the wider national and international calls for more professionals with IT competencies who can deal with the increasing challenges of digitalization, artificial intelligence, cyber security, social media, and sustainability. It is noteworthy that any list of priorities related to the development of IT always includes sustainability, but never in first place. Sustainability typically comes last or close to the bottom of the list. It is as if IT experts and researchers must first solve the challenges that were created by IT. This phenomenon is not unique to IT; it has been a long-studied issue with technology in general [1,2].

This article aims to address the future challenges in sustainability and IT by reversing the existing order of prioritization. In short, the greatest challenge that humanity will face in the coming decades is sustainability. Therefore, the question is, how can IT design, develop, and assist in maintaining the ambitious, albeit difficult to grasp, sustainability agenda?

To address this question, we first define what we mean by sustainability. Then, we illustrate four different ways in which researchers at the Department of Sustainability and Planning at Aalborg University have been creating, operating, adapting, and maintaining IT-based systems in the service of sustainability.

## 2. Sustainability

Aalborg University has committed to contributing integrally to the goals of Agenda 2030, also known as the Sustainable Development Goals or SDGs [3]. This effort is mostly focused on social and economic aspects. To complement this, the university has been officially recommending to students and staff that they contribute to the integration of this framework with that of the Stockholm Resilience Centre on planetary boundaries [4]. Therefore, for us, sustainability means addressing the SDGs within planetary boundaries (PB). However, this is still a work in progress. There is no agreed upon framework on how to systemically integrate all the 17 SDGs, with their 231 indicators, with the sciences required to monitor and hold the nine dimensions of the planetary boundaries within a safe space of operation.

Like with any other systemic approach for sustainability, this one is also clear in its scope regarding the overall systemic objective, but it lacks precision when it comes to providing tools to measure the impact of any specific project or intervention. There are two initiatives to solve this conundrum: first, the Six Transformations approach, with a broad network of collaboration which attempts to translate the SDGs and PB objectives into areas of action [5]; second, a report by the Stockholm Resilience Centre forecasting four different scenarios of action, where only one will support the achievement of SDGs within PB [6].

The Department of Sustainability and Planning at Aalborg University has been sporting this name since June 2023. Before this, it was simply called the Department of Planning. The new name is the result of years of effort to make sustainability an integral part of all research and educational activities at the university. For example, in 2008, students could study MSc Sustainable Energy Planning and Management, and in 2013, programs in MSc Sustainable Cities and MSc Sustainable Design were introduced. These programs aim at integrating sustainability as a core element in the profile of graduates and not just an add-on to complement the conventional capacities of traditional engineering, science, and technical professionals.

Likewise, research has increasingly become sustainability centered. To illustrate how this has shifted the logical balance from sustainability being just one more of the many fields of IT application to becoming an integral part of IT-supported research, we present four research areas of the Department of Sustainability and Planning in the following section: GIS (geographic information systems) for sustainability; energy planning supported in tailored IT developments; enhancing sustainability through digitalization and annually renewable resource-based components' production for the building industry; and re-assessing the role of technology, especially IT and communications, in the interaction among users, citizens, and experts in co-design processes.

## 3. Geoinformatics and Earth Observation Technologies for Sustainability

Planning for the United Nations' SDGs requires an integrated approach that considers environmental concerns along with social and economic interactions [5,7]. Monitoring developments over time and place to provide a simulated picture of future perspectives can tell us how sustainable our current practices, behaviors, and lifestyles are. Following the digital advances in the past decade, GIS data, tools, and methods are playing a crucial role in successfully supporting sustainability efforts across different domains by providing data about the environment and social and natural resources, digital tools for analyzing these data across space and time, and methods/tools for monitoring the historical development and futuristic simulation of the planet [8].

In addition, trends within the last decade have resulted in a more open, digitally fast world in which the following have occurred: (a) petabytes (if not exabytes) of historical geographical data collected through Earth Observation (EO) technologies, such as satellites and sensor networks (e.g., Copernicus and NASA programs), have become available for research, and data spaces/lakes are being introduced to the research community; (b)

the unprecedented availability of IT technologies, even on smart phones and tablets, has given rise to citizen science initiatives and data campaigns generating unique data; (c) open source- and open science-based research tools and methods have enhanced the scalability and transparency of scientific research methods and findings; (d) advancements in supercomputers, e.g., LUMI, HPC, and their advanced services, are serving the research community in diverse ways [9,10]. These advancements collectively have enabled the development of the highly complex digital twins (DT) of planet Earth (i.e., DestinE) and several thematic DTs (i.e., DT of the Oceans, DT-Biodiversity) [10]. These DTs serve as digital replicas of the Earth in the past, present, and future, which has been a game changer for sustainability-related studies. In other words, the DTs function like simulation games for the Earth, bringing together its atmosphere, biosphere, and geosphere, and enabling different stakeholders to interact with the game [11].

The Geoinformatics and Earth Observation (GEO) group embraced these open source and digital twinning trends while developing holistic data harmonization, analytical tools, and methodological approaches to address environmental and societal challenges across different application areas related to sustainability, with IT technologies, stakeholder inclusion, and designing for societal impacts being an integral part of these developments.

The contributions of the GEO group include the development of spatial decision support infrastructure, tools, and models for marine and freshwater planning. The EU Maritime Spatial Planning Directive [12] has documented the growing international need for the more holistic planning of human activities at sea. The group has targeted this need in its recent research on using data and models to better capture the land–sea interactions of substance emissions and their cumulative impact on the environment [13] and has investigated the distributional fairness of benefits [14] from ecosystem services [15] considering multiple uses [16]. Existing sustainability challenges involve the need to integrate marine and terrestrial planning while considering land–sea interactions, the water continuum from freshwater to marine waters, and the ecosystem dynamics [17]. The GEO group manages some European Union (EU) Horizon-funded projects on these challenges. For instance, a project to develop data services fit-for-purpose for integrating into DTs, to assist European research reach healthy waters across land and sea (the ongoing AquaINFRA project (2023–2026) https://aquainfra.eu/ accessed on 24 March 2024—Infrastructure for Marine and Inland Water Research). Another project starting later in 2024 (the upcoming SEADITO project (2024–2027)—Social Ecological Analysis and Models for Digital Twin Oceans) will advance social–ecological models and what-if scenarios in support of DTs, bringing intangible, qualitative data and citizens' science into the models. This is to acknowledge the importance of both space and place in geographical analyses, as outlined by Cabrera-Baron et al. (2018) [18] and Papadakis et al. (2020) [19] .

In addition to modeling scenarios for the seas, the group has assisted in the development of migration scenarios for Europe (the finalized project FUME (2019–2022) https://futuremigration.eu/ accessed on 24 March 2024—Future migration scenarios for Europe), simulating migration patterns from the origin country, e.g., Africa and the Middle East, to settlements in European urban neighborhoods, in cities such as Copenhagen and Amsterdam [20]. The IT-based modeling of such scenarios improves our understanding of the complexity, diversity, motivations, modalities, and spatial patterns of migration at multiple geographical scales and supports migration policies, integration measures, and labor market and cohesion policies at different governance levels.

Besides tools/models, infrastructure, and services for integrating data for scenario-based modeling, the group has assisted with monitoring developments already taking place in the world. For example, the group has utilized satellite data to monitor terrestrial ecosystems such as settlements, forests, water bodies, and agricultural lands. Google Earth Engine and Microsoft planetary computers have been utilized in the development of interactive tools for monitoring land cover/use changes, urbanization, deforestation, and wildfires' susceptibility mapping [21,22]. In addition, the monitoring of public health was

supported with a data-driven approach that explored the spatiotemporal patterns of the COVID-19 pandemic in Europe at different levels, making use of machine learning methods for exploring the relationship between cityscape characteristics, demographics, climatic factors, and the pandemic outbreak rate [23].

The group has an iterative approach, where platforms and tools are continuously tested, adapted, and assisted with new tools/models/scenarios. A specific example of a tool the group helped develop is the device-flexible open-source Baltic Explorer platform, visualized in Figure 1. It is a collaborative GIS, systematically visualizing marine maps in an interactive, click-on-and-off way, and combining official Baltic Sea data with local, project-based data, while enabling users to collaboratively discuss, explore, and plan the future by drawing new data, tested in workshops for local planners in 2019–2020 in Copenhagen [16], Sweden [24], and Riga [25]. Engaging users in transparent scenario-based workflows is important for increasing trust and enabling more long-term solutions when planning outcomes that are created in collaboration [26]. These are examples of the different ways in which sustainability has become an integral part of the IT developments of the GEO group, which aims to consider ecosystem and societal complexity, and open-source user needs, all in support of more informed governance. The GEO group aims to work for the continuous integration of toolsets and research environments designed for holistic, scenario-based, transparent processes to support sustainable planning. It supports open research based on the assumption that the more users know and can access tools and data, the more they can inform sustainable planning and have a greater impact, such as with the EU Horizon program.

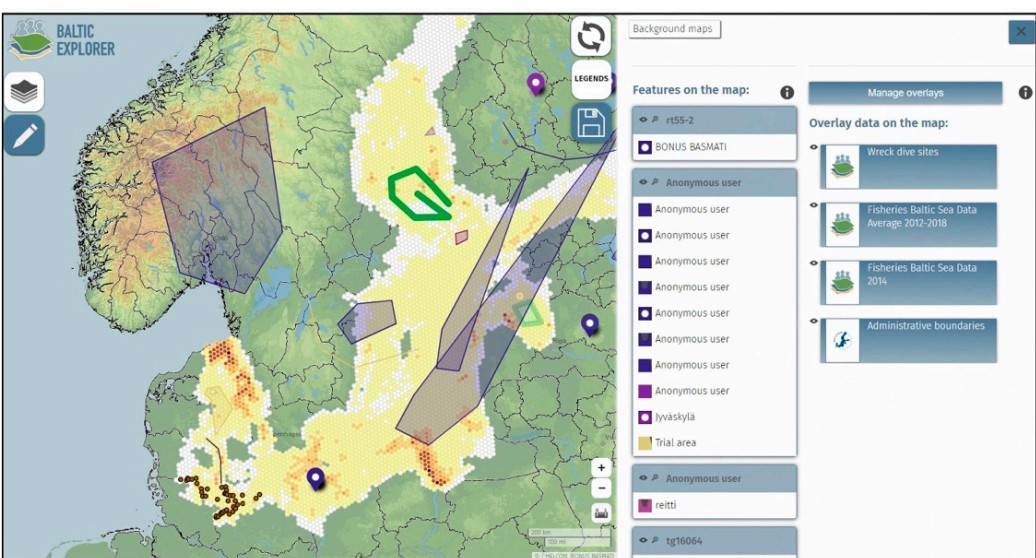

**Figure 1.** The Baltic Explorer demo with options to input one's own data, set background maps, and explore Baltic Sea data (can be adapted to other regions).

## 4. Sustainable Energy Planning

Using IT in the form of energy system models is an embedded part of work carried out by the Sustainable Energy Planning group. The group uses an interdisciplinary approach to energy planning by combining energy system modelling and geographical information systems (GIS) with socio-economic and institutional dimensions. The in-house model EnergyPLAN [27] was developed at the end of the 1990s and has been further enhanced since then. The model, which has more than 3000 users worldwide, is being actively used for teaching at AAU as well as many other universities worldwide. The tool is also being widely used by researchers and has been reportedly applied in more than 300 journal articles [28]. Figure 2 shows an input–output diagram of the model.

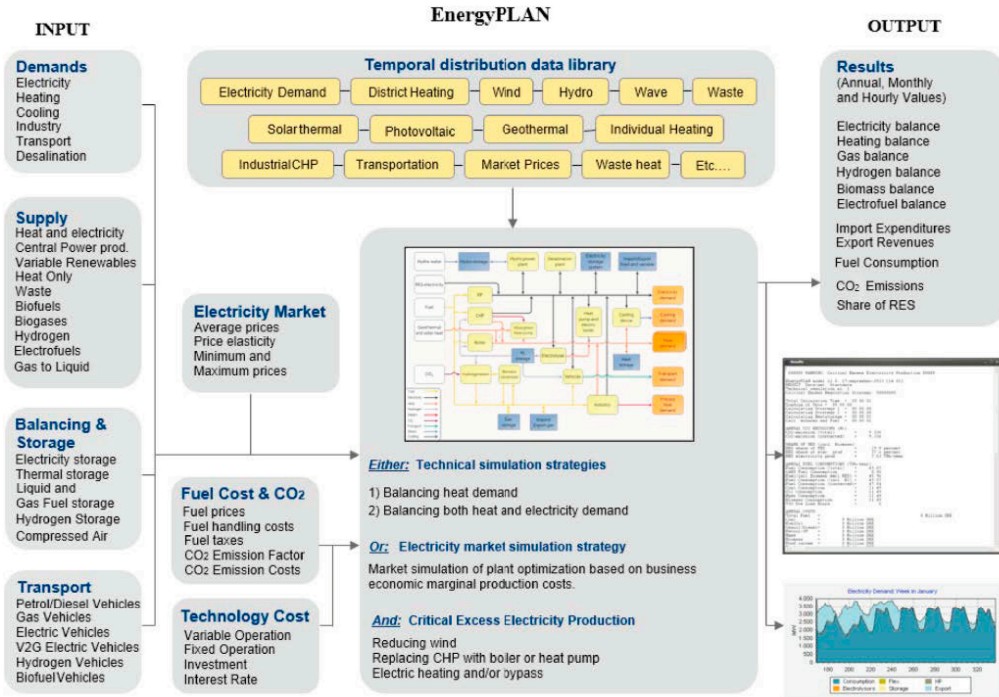

**Figure 2.** Input and output data of EnergyPLAN [27].

Sustainable and renewable energy systems are at the core of energy system modeling, as it is possible to create various scenarios that enable reaching the regulatory targets, as well as challenging the ones already in place. Scenario modeling can investigate system design that can reach 100% renewable energy levels. By using a scenario simulation tool instead of an optimization tool, it is possible to investigate a wide range of alternatives for how to achieve sustainability in energy systems [29]. Energy system modeling also enables testing concepts such as smart cities and smart grids. Previous analysis shows that it is crucial to look at the overall system instead of creating sub-optimal systems within a larger system to achieve financial and environmental sustainability [30–32].

By creating different scenarios, the type of technology and the energy efficiency measures needed in future energy systems can be tested, as at a certain point, it is more cost-effective to change the supply technology [33]. Once defined, these measures can be translated to specific IT solutions that are needed to achieve higher shares of renewable energy in the system. For example, digital solutions such as smart meters or smart charging can help integrate a higher share of renewables and flatten the peaks from consumption curves. These types of smart solutions will enable demand flexibility as we are heading toward large numbers of electric vehicles (EVs) on the roads. However, it is not only smart electricity grids that are needed to help the integration of renewable energy but also smart heating and smart gas grids [29]. All these concepts rely on digital solutions that will enable demand tracking and response. Nonetheless, the visualization of the geographical distribution of energy demand and supply and renewable energy potential is an extremely important part of energy planning and the sustainable energy transition.

Alongside the development of energy system modeling at the end of the 1990s, heat atlases of demand and supply sources were developed to help estimate the future potential of the heating sector in Denmark. This approach uses geographical analysis as part of energy planning, where energy systems and resources are analyzed by using GIS to determine the geographical constraints and distribution of renewable energy and energy system infrastructure that are important for modeling and planning. The maps are available online and represent a methodological approach to using geographic mapping for more strategic energy planning [34]. The most detailed analysis is performed as part of

the Danish Heat Atlas, where it is possible to see heat demand within urban areas at the municipal, regional, and national levels. The work has been extended to other parts of Europe with the EU Thermal Atlas [35], where the recommended DH levels are proposed, as well as the locations of excess heat sources. Creating a heat atlas enables calculations of the district heating distribution grid expenditure, to model in detail the investment costs in district heating grids. Apart from heating, GIS has been used for mapping PV panels on rooftops in Denmark, mapping $CO_2$ point sources, and analyzing $CO_2$ infrastructure by looking into the potential locations of PtX plants [36]. The geographical distribution of the foreseen changes in electricity consumption and production at different transmission grid nodes can pose a barrier to the sustainable development of energy systems [37]. The aim is always to develop a geographical model which uses a methodology that can be applied elsewhere.

## 5. Enhancing Sustainability in the Building Industry, Embracing Bio-based Materials and IT Technologies

The building industry holds a substantial responsibility for resource consumption and greenhouse gas emissions, utilizing nearly half of all extracted resources and contributing over a third of greenhouse gases [38]. The industry's heavy reliance on resources like steel, concrete, and aggregates has intensified the urgency for rapid innovation.

The BioMat initiative in the Design for Sustainability research group seeks to explore locally available, annually renewable material resources, with a particular focus on natural fiber derived from agricultural waste streams or industrial crops such as flax, hemp, and jute. Since these bio-resources are still non-conventional, diverse IT applications are essential to reach the intended sustainability goals in this field.

Computational and simulation technologies are applied in this case to establish the innovative geometries suitable for applying the new materials. The same computational technologies serve as the input for the applied digital fabrication techniques, specifically additive manufacturing, that have been pivotal in this case. In this context, the development of biocomposites or natural fiber-reinforced polymer composites (NFRPC)—hybrid materials of at least one biomass-based component, based on the combination of fibers and binders—has been a multidisciplinary effort. By aligning natural fibers to stress and deformation zones and leveraging efficient digital production methods from industries like aviation, automotive, and textiles, these biocomposites offer tunability and optimized material utilization. [39].

In different projects, a multidisciplinary approach has been introduced to achieve maximum accuracy and optimized material distribution for innovative architecture. To investigate the integration, 1:1 built mockups have been constructed to validate the entire design strategy that is based on the 'Materials as a Design Tool philosophy' [40,41], integrating (digital) fabrication parameters, design functions, and material aspects, guided by sustainable production methods and simulation technologies for building components.

To exemplify this approach and the way that IT and sustainability are integrated in this context, the Tailored Biocomposite Mockup of 2019 was created [42,43]. It was planned that the additive manufacturing technology, known as Tailored Fiber Placement (TFP), would be used to create the double-curved building element.

TFP (Tailored Fiber Placement) technology (Figure 3) is a digitally controlled embroidery process that enables the production of complex textile preforms with force flow orientation and the precise placement of natural fibers. It was adapted from the textile and aircraft construction industries to produce ultimate lightweight solutions. In the building industry, weight reduction combined with material deposition efficiency using this technique promotes enhanced sustainability and reduces both energy and resource consumption.

In this technique, long natural fibers are incorporated into the binding matrix in a targeted manner along the main load path, so that their tensile strength properties can be optimized. The building component created was a human-sized canopy in the form of a

single-curved, free-standing structural component, generatively designed as a lightweight structure, 225 cm high and 125 cm wide and weighing 35 kg, with five yarns of flax fibers and reinforced by non-woven flax fibers (Figure 4).

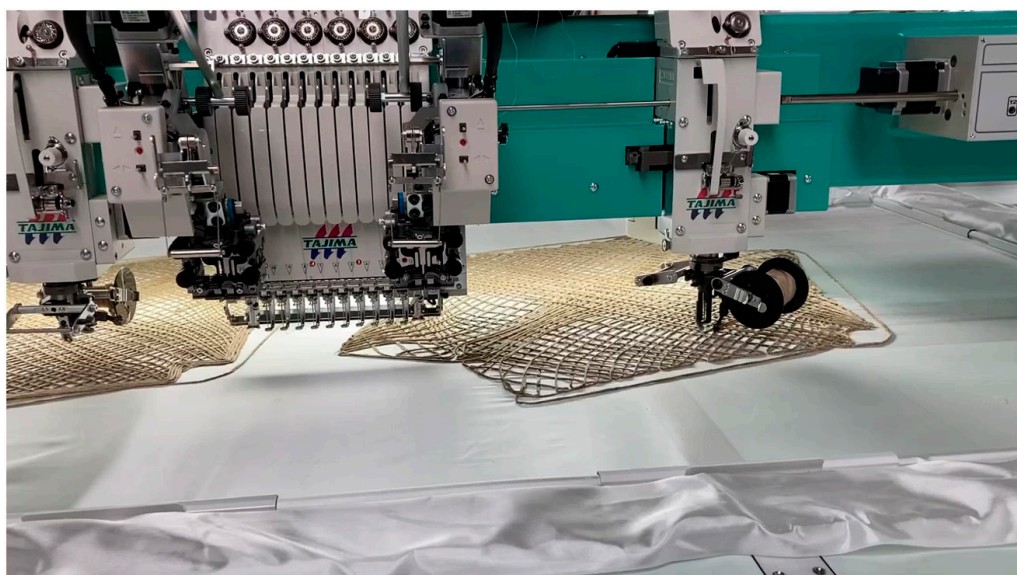

**Figure 3.** Tailored Fiber Placement (TFP) of flax fibers. ©BioMat.

The initial geometry was created using Rhinoceros plugins Grasshopper, Galapagos, and Millipede, where the geometry performing the smallest deformation was chosen. The selected design was later topologically optimized to reduce the amount of unnecessary material.

The parameters from the topology optimizations that took place using MATLAB were later used as constraints for an agent-based system (code in processing using a Plethora plugin) to simulate the optimum smooth direction of the continuous flax fibers. The different tracks coming from individual agents were post-processed into the mesh geometry, and the most feasible design was selected accordingly. The line patterns correspond to the force flows and hence indicate the fiber paths in the canopy structure.

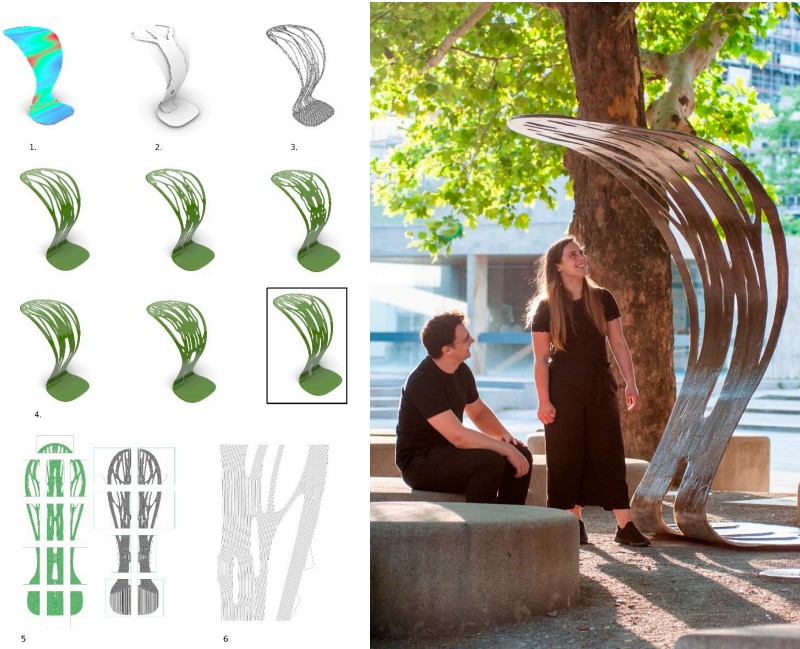

**Figure 4.** (**Left**): Design process: (1) basic shell optimization, (2) topology optimization of a single-curved canopy, (3) generated structure by agents attracted by results from topology optimization,

(4) selection of the final design from generated variants, (5) CAD paths of individual preforms, (6) details of path for tailoring. (**Right**): The 1:1 lightweight TFP free-standing building component ©BioMat.

In order for the TFP embroidery machine to be able to join the predefined fiber patterns into textile preforms, the previously determined 3D form of the canopy was unwound and transferred onto a level surface. The machine software "WIREpath" (https://www.filacon.com/fiber (accessed on 24 March 2024)) was applied for performing the process of transferring the CAD (computer-aided design) data to the machine and setting diverse stitching-related parameters of the flattened geometrical templates, to be prepared for the stitching process.

Most of the fiber-reinforced polymer composite products in which such a fiber placement method is used require a counter-mold. For this mockup, a reusable mold was created from components of flexible CNC-milled balsa timber. The finished textile preforms were placed onto the prepared mold and then impregnated with an epoxy resin and further formed and hardened in a closed vacuum-assisted process.

The presented mockup is an example of the successful application of the automated TFP technique with long natural fibers, integrating diverse IT technologies and simulation techniques to support the targeted sustainable development of the building industry development scheme. This method is well-suited for the fabrication of precise shell and panel structures with controlled fiber orientation along with the predominant tensile forces. In additional work, building components were developed using TFP technologies, while eliminating the necessity of applying large-sized molds to achieve greater ecological impact [44,45].

## 6. Techno-Anthropology for Sustainable Technologies and Engineering

Concerns about making IT sustainable and developing it in the service of sustainable practices are supported by efforts to integrate epistemological, ontological, and normative perspectives while performing IT research. This is particularly important when most IT research, including the research carried out at universities, is determined by commercial needs and priorities. Researchers at the Techno-Anthropology and Participation Research Group are working on this issue. With the concept of Techno-Anthropology (T-A), they are promoting methodological approaches for the design and assessment of digital technologies that innovate epistemologically, ontologically, and normatively. More specifically, these researchers acknowledge that the relationship between IT and sustainability is deeply socio-ecological and requires new methods and approaches to spotlight the relationship between the social, ecological, and technological aspects of the design, implementation, and adoption of digital technologies.

To accomplish this, researchers in T-A consider technologies that remain between stakeholders, technical experts, and technical artifacts. By adopting a normative stance toward sustainable, democratic, and just societies, T-A researchers have been able to engage in projects dealing with the methods and approaches to design and have implemented and assessed digital technologies with the explicit involvement of the actors—lay people and experts—who are likely to be affected by the technologies themselves. In this way, not only do the technologies become more sustainable but also the engineering practices place greater attention on their ecological, ethical, social, and political implications.

Engineering and technology projects are normally driven by technical experts (and an underpinning expert logic) with commercial purposes. T-A projects contrast this *modus operandi* and typically involve not only experts but also lay people, such as employees or local citizens impacted by IT research. T-A projects position IT and sustainability in an inclusive way, privileging participatory design and giving voice to those who are normally overlooked. Inclusion takes different aspects in a T-A perspective, from facing inequalities to biodiversity monitoring and touching upon workers' well-being and autonomy in relation to digital tools.

For example, through the participatory design of a digital platform, commonfare.net, T-A researchers have worked on translating complex concepts like the one of Commonfare, or the welfare of the common, into IT systems [46]. To further question socio-economic inequalities, when working on commonfare.net, researchers gave visibility and support to activist groups, implemented means for the creation of alternative, group-based currencies [47], and rethought the way that trust could be visualized when dealing with common-based dynamics [48]. As participatory design, the work conducted on Commonfare has exemplified the repoliticization of participatory design practices in the last decade [49] and has brought into question the role that design researchers play between grassroots initiatives and institutions [50].

When dealing with biodiversity monitoring, inclusion takes two different perspectives, exemplified by one of the recent projects T-A researchers have been involved in—OBAMA-Next (Observing and Mapping Marine Ecosystems—Next Generation Tools, 2022–2026). In this case, T-A researchers have two main roles. First, in line with the tradition of participatory design, T-A researchers have been facilitating the relationship between experts in biodiversity monitoring and the IT developers responsible for the design and implementation of new technologies for biodiversity monitoring. Second, through an analysis of existing citizen science initiatives, T-A researchers have been promoting a democratization of knowledge production in relation to biodiversity monitoring.

When dealing with workers' autonomy and well-being, T-A research is engaged in a Europe-wide collaboration that focuses on the competences needed by design researchers to play a role in facing the present socio-ecological challenges [51]. In this case, T-A researchers are contributing via a focus on platform cooperatives, in which the economic possibilities afforded by digital platforms are seen as subordinated to the interests of the workers involved. By researching the cooperative form of enterprise in relation to digital platforms, T-A researchers are contributing to subordinate IT design, development, and implementation, to the quest for sustainability, as well as to the organization of economic process. Another example is the ongoing work conducted on the implementation of PtX-infrastructure at the port of Rønne on Bornholm Island (Denmark). This research involves both local communities and port workers mapping their assessment of the PtX-infrastructure. The normative foundation aims to establish a project that is supported by port workers, local citizens, and other stakeholders who otherwise might be left with no voice in the design of the Danish energy infrastructure. The findings can inform management in the design of the infrastructure to avoid emerging ethical dilemmas and controversies following the green transition.

If OBAMA-Next and PtX-infrastructure are ongoing projects, Commonfare was successful not only academically but also in developing pieces of technologies based on new assumptions. Moreover, all these research efforts have found their way into education and into the reflection on ethics in engineering. For example, Børsen and Chance provide an overview of state-of-the-art research on engineering ethics education [52], focusing on experts in different areas of engineering and how they can integrate ethical, participatory, and ecological reflections in their teaching, research, advice, and innovation activities. To ensure that IT projects start with ecological and sustainability objectives and then follow up with appropriate technology designs, it is imperative to redesign engineering education to embrace normative reflections. This is what this project contributes to. In conclusion, techno-anthropological research contributes explicitly to developing socio-ecologically sustainable IT, in design, development, implementation, and assessment.

## 7. Conclusions

In this article, we have presented a sample of the type of research conducted at the Department of Sustainability and Planning at Aalborg University. This department is unconventional in the sense that its research groups are defined by societal challenges. This is a contrast to traditional disciplines which, in many instances, are defined by technologies. The fact that it was recently renamed the Department of Sustainability and Planning

reflects the efforts of all the research groups involved to improve their focus on solving sustainability challenges.

We have presented the GEO group and how it applies IT technologies and developments in its available sets of data and computational capacity. This effort aims to map and monitor water availability, water levels, and other physical entities in time and space. This is key to consolidating knowledge on how human activity affects ecosystems and how they evolve and to encourage and guide research towards planning outcomes where human impacts remain within planetary boundaries. The GEO group has also developed the capacity to monitor the distribution of resources to immigrant populations, and to monitor the health of citizens in different parts of the world. This is also necessary to support the action to achieve the Agenda 2030 goals. All these efforts require not only the application of IT to sustainability but also the innovation and adaptation of IT knowledge and tools.

The research on energy planning concentrates not only on technological development—the focus of private and funding bodies all over the world—but also on refining the planning and design of a complete sustainable energy system. It departs from the realization that such a system will not be able to exist unless many different production, distribution, and storage technologies are developed, optimized, and integrated. This is a supranational effort that requires a high degree of coordination across territories, political boundaries, and levels of government by using a framework of strategic energy planning [53] . Fulfilling this framework is possible by applying and developing IT tools.

The BioMat initiative of the Design for Sustainability research group has been concentrating on innovation and sustainability in the building industry depending on applying non-conventional annually renewable materials integrating diverse manufacturing methods. In the future, construction will need to move away from conventional energy-intensive materials. BioMat has been experimenting with diverse new materials that will not only replace energy-intensive conventional materials but also perform better. This requires, as illustrated, IT-based innovative methods to produce the material from renewable and waste sources, living up to the principle of decoupling and circularity, which is fundamental to achieving sustainability [5] . Additionally, the materials' geometry and manufacturing are optimized for the use of a variety of IT programs and tools.

Researchers in techno-anthropology focus on understanding IT and digital technologies and organizing IT design, development, implementation, and assessment. IT and digital technologies can also be designed for the service of the commons. This aims to enhance public participation and to allow for citizen involvement in the creation of the same technological systems that promise to support the achievement of the sustainability agenda. This requires, as illustrated, adjustments in the technologies and tools available and under development, which, in turn, requires innovation about the interaction between humans and technology.

Our goal has been to collect, organize, and present examples of research which is defined by sustainability priorities and includes, adapts, and innovates on IT. This effort has many limitations which we want to express in terms of desirable research efforts in the years to come. Future research is needed to understand how these initiatives compare to other similar endeavors in other institutions around the world; to assess the impact of this research in relation to Agenda 2030, not as a superficial reporting effort, which is already done [54], but in a rigorous and scientifically measurable way; and finally, to relate our particular take to the robust discussions on the current and future meanings of sustainability in general and within the IT research community.

Recently, the Technical Faculty of IT and Design at Aalborg University has begun the ambitious project of creating several new IT educational programs. IT and software experts have expressed their frustration because, in their experience, such programs are not attractive to female students. We contend that this happens because they are being developed around technology, which historically has produced many disciplines that are male-dominated [55]. In line with the argument of this paper, some of the authors have tried to

suggest a different approach called Sustainability and IT, where the emphasis is on the challenges to be addressed, in line with the department's research tradition and educational programs—for example, Sustainable Design, Sustainable Cities, and Techno-Anthropology—which exhibit gender balance and are inclusive in other ways too. This initiative has not been prioritized on the grounds that students need to first master the technology, and only after that, its applications and social significance, also in relation to sustainability.

We present this as our concluding reflection, because the variety of research presented in this article also reflects our capacity to contribute to the sustainability agenda. One of the main challenges addressed by Agenda 2030 is gender inequality at all levels, in all societies. What we have witnessed inside our own institution is that by focusing on technology in new IT programs, we risk diminishing our capacity to contribute to sustainability from the outset. If we end up with several educational programs that primarily attract male students, we would have failed to be inclusive, therefore failing from the start to contribute to the SDGs. In addition, placing SDGs as a central focus for IT innovation enables educational and societal inclusiveness beyond only gender balances.

**Author Contributions:** Writing—original draft, A.F.V.P., I.R.S., H.D., J.J.A., I.M.B., T.B. and M.T.; Writing—review & editing, A.F.V.P. All authors have read and agreed to the published version of the manuscript.

**Funding:** The upcoming SEADITO project, the ongoing AquaINFRA project, and the finalized FUME project were made possible due to funding from Horizon Programs on Research and Innovation and Research Infrastructures from the European Union in the period 2020–2023. The finalized BONUS BASMATI project received funding from BONUS (art. 185), call number 2015-77, funded jointly by the EU, Innovation Fund Denmark, Swedish Research Council, Formas, Academy of Finland, Latvian Ministry of Education, and Science and Forschungszentrum Julich GmbH (Germany). The OBAMA-NEXT project has been approved under HORIZON-CL6-2022-BIODIV-01-01: Observing and mapping biodiversity and ecosystems, with grant agreement No. 101081642.

**Data Availability Statement:** The data presented in this study are available on request from the corresponding author.

**Conflicts of Interest:** The authors declare no conflict of interest.

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
