# Peer review of "Sustainability Meets Information Technologies: Recent Developments and Future Perspectives"

_sustainability, doi:10.3390/su16114499_

Round 1
Reviewer 1 Report
Comments and Suggestions for Authors
The manuscript titled "Sustainability Meets Information Technologies: Recent Developments and Future Perspectives" provides an insightful exploration into the integration of sustainability concepts with information technology practices. This paper successfully identifies and discusses the emergent challenges and developments in the field, proposing forward-thinking perspectives that could influence future research directions.
The paper’s premise to reverse the conventional prioritization of IT objectives towards a sustainability-first approach is both innovative and timely. This re-prioritization could potentially contribute significantly to literature and industry practices. The discussion about integrating sustainability with IT education and research is articulated clearly, with comprehensive coverage of potential impacts and benefits. This provides a solid theoretical foundation for the proposed changes. By examining case studies and current practices at Aalborg University, the paper effectively bridges theoretical concepts with practical implementations, enhancing its relevance to both academics and practitioners.
Comments on the Quality of English LanguageThe paper needs minor revision through careful proofreading to enhance readability and flow. For example (not limited to),
· "This article aims at addressing the future challenges" could be revised to "This article aims to address the future challenges..."
· The use of complex and lengthy sentences in sections such as the introduction (Lines 20-35) could be simplified to improve clarity and reader engagement.
· “reserch” should be corrected to “research” (Keywords line 17).
Author Response
Dear Editor and Reviewers,
First of all, we are inmensely grateful for the set of valuable comments you generously put together for our article. It has helped us understand the many ways in which we can improve our argument. Thanks a lot.
WE HAVE INCLUDED OUR ANSWER TO ALL SUGGESTED REVIEWS IN CAPITALS AFTER THE SUGGESTIONS. WE HOPE THIS IS OK.
Reviewer 1
Requires minor editing
Comments and Suggestions for Authors
The manuscript titled "Sustainability Meets Information Technologies: Recent Developments and Future Perspectives" provides an insightful exploration into the integration of sustainability concepts with information technology practices. This paper successfully identifies and discusses the emergent challenges and developments in the field, proposing forward-thinking perspectives that could influence future research directions.
The paper’s premise to reverse the conventional prioritization of IT objectives towards a sustainability-first approach is both innovative and timely. This re-prioritization could potentially contribute significantly to literature and industry practices. The discussion about integrating sustainability with IT education and research is articulated clearly, with comprehensive coverage of potential impacts and benefits. This provides a solid theoretical foundation for the proposed changes. By examining case studies and current practices at Aalborg University, the paper effectively bridges theoretical concepts with practical implementations, enhancing its relevance to both academics and practitioners.
THANKS FOR THE NICE SUMMARY. IT IS PRECISELY WHAT WE INTEND.
Comments on the Quality of English Language
The paper needs minor revision through careful proofreading to enhance readability and flow. For example (not limited to),
- "This article aims at addressing the future challenges" could be revised to "This article aims to address the future challenges..."
WE HAVE CHECKED WITH AN EXPERT AND BOTH FORMS ARE CORRECT. WE STICK TO THE ORIGINAL.
- The use of complex and lengthy sentences in sections such as the introduction (Lines 20-35) could be simplified to improve clarity and reader engagement.
THANKS FOR THE SUGGESTION. THE ARTICLE WAS PROOF READ BY AN EXPERT BEFORE SUBMISSION. WE HAVE RE-REVISED IT IN ITS ENTIRETY WITH FOCUS ON SHORTENING LONG CONVOLUTED SENTENCES.
- “reserch” should be corrected to “research” (Keywords line 17).
WE ARE AFRAID THIS LINE WAS NOT INCLUDED IN THE DRAFT FOR CORRECTIONS WE GOT, BUT WE HAVE MADE SURE THAT THE WORD RESEARCH IS CORRECTLY SPELLED EVERYWHERE. THANKS.
Reviewer 2 Report
Comments and Suggestions for Authors
Brief summary
This article is an initiative by Aalborg University's Technical Faculty of IT and Design to pivot IT education and research towards sustainability. The university integrates sustainability deeply into its education and research including public administration, industry, civil society, Geoinformatics, Earth Observation Technologies, and infrastructure as a scope. Major contributions include the development of IT-supported systems for sustainable practices, the development of spatial decision support infrastructure, tools, and models for marine and freshwater planning, simulation migration patterns. The paper's strength is its approach which combines IT innovations with sustainable development goals (SDGs), and enhances the social and ecological applications of technology.
Weaknesses
Hypotheses: the paper does not provide specific, measurable objectives or hypotheses that can be empirically tested. This can make it difficult to assess the impact of the paper.
Scope
While a broad scope can be a strength, the paper has challenges in maintaining depth and focus. The article covers a wide array of topics from educational programs to specific technologies which makes it hard to conclude their effectiveness.
Completeness
Some missing areas can be considered to improve the article:
-No global context: the article focuses on initiatives at Aalborg University. Comparing and contrasting with similar work at other institutions worldwide would provide a broader perspective on the challenges and innovative solutions found in different cultural and regulatory environments.
-No long-term impact was discussed.
-No details on their GEO group which has also developed the capacity to monitor the distribution of resources to immigrant populations, and to monitor the health of citizens in different parts of the world.
-No details on their research on energy planning.
The paper presents a lot of valuable initiatives as a concluding summary but they are not referenced in the text. That makes the paper hard to follow and evaluate.
Relevance of topics
The topic is relevant: IT systems and technologies play a crucial role in modern societies, its impact on environmental and social systems is high. By including sustainability in IT, the article addresses a valid need for present and future technology professionals who can design systems with minimal negative environmental impact and enhanced social value.
There is a growing demand in the technology sector for sustainable practices. The regulatory environment also acts in the same direction.
References
There are 49 references. They cite mostly recent publications (within the last 5 years) and they are relevant.
Some suggestions: Please check
- - ref. [7] (line 467)
- - „The maps are available online and represent a methodological approach to using geographic mapping for more strategic energy planning (Welcome to EnergyMaps, n.d.).” (lines 207-208) (I could not reach that webpage)
- Figure 4 is not referenced in the text. The text “The design process is sketched in Figure 3.” (Line 275) is wrong, Fig 4 deals with the design process.
Both Fig 3 and Fig 4 are not too relevant to the topic
Clarity
The manuscript is clear. Its specific focus is presented, its aims should be clarified better. The manuscript is well-structured, It presents an introduction to the motivations and background, followed by detailed discussions on the various initiatives undertaken by the university. This approach is effective for readers to understand. It provides a comprehensive view of the university’s efforts.
Potential Areas for Improvement:
Section 5 Enhancing Sustainability in the Building Industry, Embracing Bio-based Materials and IT Technologies should be revised. The title is generic; however, its content is specific to Tailored Fiber Placement.
Section 6 the concept of Techno-Anthroplogy provides some examples but does not evaluate them: are they successful? Did they reach their aim? Are the best practices to follow? Is there a way to improve them?
Lack of practical impacts. I suggest including either more empirical data and more detailed case studies. This would help in demonstrating the practical impacts of the initiatives discussed.
Integrating more comparative analysis with similar initiatives at other institutions could also enhance its relevance and provide a broader perspective on the challenges and solutions in integrating sustainability in IT advancements.
The manuscript should acknowledge its own limitations, whether they be in scope, methodology, or applicability of the findings. It should also suggest areas for further research.
Future perspectives are not discussed
Conclusions drawn
The conclusions should address the ability to generalize the findings. The conclusions should directly refer to the research aims stated at the beginning of the manuscript. The conclusions should clearly reflect outcomes related to this goal based on the facts gathered through the study.
I wonder whether there can be measures that aim to measure the impact of the achievements similar to Energy systems (Line 188)
Comments on the Quality of English Language
list of errors, typos:
Line 8 there has been a greater focus
Line 17 reserch
Line 22 IT centered -> "IT-centered"
Line 53 "approach for sustainability"
Line 62 missing comma in "Before this it was simply"
Line 251 in -> into
Line 351 As a participatory design
Line 421 energy intensive -> energy-intensive
Author Response
Dear Editor and Reviewers,
First of all, we are inmensely grateful for the set of valuable comments you generously put together for our article. It has helped us understand the many ways in which we can improve our argument. Thanks a lot.
WE HAVE INCLUDED OUR ANSWER TO ALL SUGGESTED REVIEWS IN CAPITALS AFTER THE SUGGESTIONS. WE HOPE THIS IS OK.
Reviewer 2
Requires minor editing
Comments and Suggestions for Authors
Brief summary
This article is an initiative by Aalborg University's Technical Faculty of IT and Design to pivot IT education and research towards sustainability. The university integrates sustainability deeply into its education and research including public administration, industry, civil society, Geoinformatics, Earth Observation Technologies, and infrastructure as a scope. Major contributions include the development of IT-supported systems for sustainable practices, the development of spatial decision support infrastructure, tools, and models for marine and freshwater planning, simulation migration patterns. The paper's strength is its approach which combines IT innovations with sustainable development goals (SDGs), and enhances the social and ecological applications of technology.
Weaknesses
Hypotheses: the paper does not provide specific, measurable objectives or hypotheses that can be empirically tested. This can make it difficult to assess the impact of the paper.
THANKS FOR THIS CRITICAL COMMENT. AS INDICATED BY REVIEWER ONE OUR CONTRIBUTION IS THAT REVERSING THE CONVENTIONAL PRIOTIZATION OF IT AND SUSTAINABILITY CAN BE MORE APPROPIATE AND PRODUCTIVE TO RESEARCH. AND WE ILLUSTRATE THIS THROUGH THE CASE OF THE SUSTAIANBILITY AND PLANNING DEPARTMENT AT AALBORG UNIVERSITY.
Scope
While a broad scope can be a strength, the paper has challenges in maintaining depth and focus. The article covers a wide array of topics from educational programs to specific technologies which makes it hard to conclude their effectiveness.
THANKS FOR INDICATING THIS. THE ARTICLE HAS A WIDE SCOPE BECAUSE THE INTENTION IS TO ILLUSTRATE OUR PREMISE WITH EXAMPLES ON HOW WE FRAME SUSTAINABILITY AND IT IN RESEARCH AND EDUCATION. WE DON’T CONSIDER HOW TO ASSESS ITS EFFECTIVENESS. THIS SHOULD BE SUBJECT OF ANOTHER ARTICLE. WE HAVE INDICATED THIS IN THE CONCLUSION AND SUGGEST IT AS A FUTURE AVENUE OF RESEARCH. THANKS AGAIN.
Completeness
Some missing areas can be considered to improve the article:
-No global context: the article focuses on initiatives at Aalborg University. Comparing and contrasting with similar work at other institutions worldwide would provide a broader perspective on the challenges and innovative solutions found in different cultural and regulatory environments.
THIS IS OUT OF THE SCOPE OF THE ARTICLE. AND FOR SURE SHOULD BE CONSIDERED IN FUTURE RESEARCH. WE HAVE INDICATED THIS IN THE CONCLUSION AND SUGGEST IT AS A FUTURE AVENUE OF RESEARCH.
-No long-term impact was discussed.
THIS IS OUT OF THE SCOPE OF THE ARTICLE. AND FOR SURE SHOULD BE CONSIDERED IN FUTURE RESEARCH. WE HAVE INDICATED THIS IN THE CONCLUSION AND SUGGEST IT AS A FUTURE AVENUE OF RESEARCH. THANKS AGAIN.
-No details on their GEO group which has also developed the capacity to monitor the distribution of resources to immigrant populations, and to monitor the health of citizens in different parts of the world.
THE PRESENT ARTICLE IS NOT ABOUT THE CONSTITUTION OF THE RESEARCH GROUPS, BUT ABOUT THE TYPE OF RESEARCH THEY DO. WE HAVE INCLUDED LINKS TO THE WEBSITES OF CURRECT RESEARCH IN THE SPECIFIC CASE OF GEO. WE CONSIDER THIS ENOUGH FOR THE PURPOSE OF THE ARTICLE.
-No details on their research on energy planning.
WE HAVE ADDED ADDITIONAL SENTENCE TO CAPTURE THE APPROACH GROUP HAS:
“The group uses interdisciplinary approach to energy planning by combining energy system modelling and geographical information systems (GIS) with socio-economic and institutional dimensions.”
The paper presents a lot of valuable initiatives as a concluding summary but they are not referenced in the text. That makes the paper hard to follow and evaluate.
WE HAVE REVISED THE CONCLUSION AND CONSIDER THAT IT FITS OUR ARTICLE. WE SUM UP THE ASPECTS IN WHICH EACH RESEARCH GROUP IS WORKING ON SUSTAINABILITY AND IT, WE QUALIFY THE LIMITATION OF OUR ANALYSIS AND STATE FUTURE AVENUES OF RESEARCH AND WE FINISH WITH A SMALL NOTE ON GENDER EQUALITY AND EDUCATION, WHICH SHOULD ALSO BE ADDRESSED WITH MORE RESSOURCES IN ANOTHER FUTURE ARTICLE.
Relevance of topics
The topic is relevant: IT systems and technologies play a crucial role in modern societies, its impact on environmental and social systems is high. By including sustainability in IT, the article addresses a valid need for present and future technology professionals who can design systems with minimal negative environmental impact and enhanced social value.
There is a growing demand in the technology sector for sustainable practices. The regulatory environment also acts in the same direction.
THANKS.
References
There are 49 references. They cite mostly recent publications (within the last 5 years) and they are relevant.
YES! ALL RECENT STUFF.
Some suggestions: Please check
- - ref. [7] (line 467)
THANKS FOR INDICATING THIS. IT IS THE DOI OF REFERENCE 6, NOT AN ADDITIONAL REFERENCE. IT IS CORRECTED NOW.
- - „The maps are available online and represent a methodological approach to using geographic mapping for more strategic energy planning (Welcome to EnergyMaps, n.d.).” (lines 207-208) (I could not reach that webpage)
THE LINK INDICATED IN THE REFERENCES (NOW 46) WORKS. IT IS https://www.energymaps.eu/
- Figure 4 is not referenced in the text. The text “The design process is sketched in Figure 3.” (Line 275) is wrong, Fig 4 deals with the design process.
Both Fig 3 and Fig 4 are not too relevant to the topic
THANKS FOR POINTING THIS OUT. WE HAVE REVISED THE FIGURES AND CONSIDER THAT THEY ALL CONTRIBUTE AND ILLUSTRATE OUR POINTS. WE CHECKED THAT THEY ARE ALL MENTIONED IN THE TEXT. THANKS AGAIN.
Clarity
The manuscript is clear. Its specific focus is presented, its aims should be clarified better. The manuscript is well-structured, It presents an introduction to the motivations and background, followed by detailed discussions on the various initiatives undertaken by the university. This approach is effective for readers to understand. It provides a comprehensive view of the university’s efforts.
THANKS. WE HAVE IMPROVED THE STATED SCOPE OF THE PAPER.
Potential Areas for Improvement:
Section 5 Enhancing Sustainability in the Building Industry, Embracing Bio-based Materials and IT Technologies should be revised. The title is generic; however, its content is specific to Tailored Fiber Placement.
IT IS STATED THAT THE TFP MACHINE IS AN EXAMPLE OF THE APPROACH.
Section 6 the concept of Techno-Anthroplogy provides some examples but does not evaluate them: are they successful? Did they reach their aim? Are the best practices to follow? Is there a way to improve them?
AS STATED IN THE TEXT WHAT IS NOVEL IS THE APPROACH. THE EVALUATION OF SUCH INITIATIVES WILL BE COVEREDE IN FUTURE ARTICLES AS WE HAVE NOW STATED IN THE CONCLUSION.
Lack of practical impacts. I suggest including either more empirical data and more detailed case studies. This would help in demonstrating the practical impacts of the initiatives discussed.
WE HAVE INCLUDED SEVERAL EXAMPLES OF FOUR RESEARCH AREAS. WE CONSIDER THIS ENOUGH FOR THE PURPOSE OF THE ARTICLE. FOR SURE THERE WILL BE MORE ACADEMIC PRODUCTION ASSESSING IMPACTS AND EVALUATING THE STRENGHTS OF THESE INITIATIVES, BUT THESE FALLS OUTSIDE THE SCOPE OF THIS ARTICLE. WE POINTED THIS OUT AS A LIMITATION AND FUTURE AVENUE OF RESEARCH.
Integrating more comparative analysis with similar initiatives at other institutions could also enhance its relevance and provide a broader perspective on the challenges and solutions in integrating sustainability in IT advancements.
YES. AND WE LOOK FORWARD TO HAVING THE OPPORTUNITY TO DO THIS. BUT UNFORTUNATELY IT FALLS OUTSIDE THE SCOPE OF THIS PAPER.
The manuscript should acknowledge its own limitations, whether they be in scope, methodology, or applicability of the findings. It should also suggest areas for further research.
WE HAVE ADDRESSED THIS IN THE CONCLUSION THANKS.
Future perspectives are not discussed
WE HAVE ADDRESSED THIS IN THE CONCLUSION THANKS.
Conclusions drawn
The conclusions should address the ability to generalize the findings. The conclusions should directly refer to the research aims stated at the beginning of the manuscript. The conclusions should clearly reflect outcomes related to this goal based on the facts gathered through the study.
WE HAVE ADDRESSED THIS IN THE CONCLUSION THANKS.
I wonder whether there can be measures that aim to measure the impact of the achievements similar to Energy systems (Line 188)
WE HAVE ADDRESSED THIS IN THE CONCLUSION THANKS.
Reviewer 3 Report
Comments and Suggestions for Authors
In Geoinformatics, is important to consider not only issues regarding space, but also place. The operationalization of place through GIS is still a challenge, especially when we are focused in offering specific sustainable solutions to improve people´s quality of life. The article needs to arise this question. I suggest to include in your references the following investigations:
https://www.mdpi.com/2413-8851/2/3/73
https://www.tandfonline.com/doi/full/10.1080/15230406.2019.1598894
Please add a schematic Figure (like Figure 2) to summarize sections 3 to 6?
I consider the SDG as important framework for sustainability. However, the SDG are not obligatory for national and local governments. Please add some discussion about this issue.
Please expand your discussion beyond the objectives of Aalborg University. Additionally, include more information regarding other sustainability-technlogy frameworks, and contrast these frameworks with your proposal.
Comments on the Quality of English LanguageMinor editing of English language required
Author Response
Dear Editor and Reviewers,
First of all, we are inmensely grateful for the set of valuable comments you generously put together for our article. It has helped us understand the many ways in which we can improve our argument. Thanks a lot.
WE HAVE INCLUDED OUR ANSWER TO ALL SUGGESTED REVIEWS IN CAPITALS AFTER THE SUGGESTIONS. WE HOPE THIS IS OK.
Reviewer 3
Requires minor editing
Comments and Suggestions for Authors
In Geoinformatics, is important to consider not only issues regarding space, but also place. The operationalization of place through GIS is still a challenge, especially when we are focused in offering specific sustainable solutions to improve people´s quality of life. The article needs to arise this question. I suggest to include in your references the following investigations:
https://www.mdpi.com/2413-8851/2/3/73
https://www.tandfonline.com/doi/full/10.1080/15230406.2019.1598894
THANK YOU FOR POINITNG THIS OUT. WE HAVE INCLUDED A PHRASE IN PAGE 3 REFLECTING UPON THE IMPORTANCE OF SPACE VS PLACE IN GEOGRAPHICAL ANALYSIS AND CITED THE INDICATED PAPERS.
Please add a schematic Figure (like Figure 2) to summarize sections 3 to 6?
WE HAVE USED FIGURES THAT ILLUSTRATE THE RESEARCH AND HAVE NOT DEVELOPED A FLOW DIAGRAM OR ANYTHING SIMILAR TO REFLECT THE CONTENT OF THIS ANALYSIS. IT IS A GOOD SUGGESTION, BUT WE CANNOT MEET IT WITH THE TIMEFRAME FOR THE REVISION.
I consider the SDG as important framework for sustainability. However, the SDG are not obligatory for national and local governments. Please add some discussion about this issue.
WE ARE AWARE THAT THERE IS A RICH AND URGENT ACADEMIC DEBATE ABOUT WHAT IS SUSTAINABILITY AND WHICH FRAMEWORKS EXIST, WHICH SHOULD BE APPROPIATE AND SO ON. WE HAVE DECIDED TO FOCUS ON THE SDGS BECAUSE THAT IS WHAT AALBORG UNIVERSITY AND DENMARK HAS DECIDED TO COMMIT TO OFFICIALLY. ADDITIONALLY, IT IS WIDELY KNOWN AROUND THE WORLD, ALTHOUGH WE ARE AWARE THAT MANY COUNTRIES, PARTICULARLY THE US DO NOT SUBSCRIBE TO THEM. A MORE ROBUST DISCUSSION ON SUSTAINABILITY IS NEEDED AND WE HOPE TO TAKE THAT UP IN FUTURE WORK.
Please expand your discussion beyond the objectives of Aalborg University. Additionally, include more information regarding other sustainability-technlogy frameworks, and contrast these frameworks with your proposal.
THIS IS A FAIR CRITIQUE, BUT UNFORTUNATELY IT FALLS OUTSIDE THE SCOPE OF THIS ARTICLE. WE POINTED THIS OUT AS A LIMITATION AND FUTURE AVENUE OF RESEARCH.
Comments on the Quality of English Language
WE HAD THE MANUSCRIPT REVISED ONCE MORE BY AND EXPERT.
Minor editing of English language required
THANKS.
Round 2
Reviewer 3 Report
Comments and Suggestions for Authors
Authors have addressed all my comments